# GATED CONVNETS FOR LETTER-BASED ASR

## ABSTRACT

In this paper we introduce a new speech recognition system, leveraging a simple letter-based ConvNet acoustic model. The acoustic model requires only audio transcription for training – no alignment annotations, nor any forced alignment step is needed. At inference, our decoder takes only a word list and a language model, and is fed with letter scores from the acoustic model – no phonetic word lexicon is needed. Key ingredients for the acoustic model are Gated Linear Units and high dropout. We show near state-of-the-art results in word error rate on the LibriSpeech corpus (Panayotov et al., 2015) with MFSC features, both on the CLEAN and OTHER configurations.

## 1 INTRODUCTION

Top speech recognition systems are either complicated pipelines or using more data that is publicly available. We set out to show that it is possible to train a nearly state of the art speech recognition system for read speech, with a public dataset (LibriSpeech), on a GPU-equipped workstation. Thus, we present an end-to-end system for speech recognition, going from Mel-Frequency Spectral Coefficients (MFSCs) to the transcription in words. The acoustic model is trained using letters (graphemes) directly, which take out the need for an intermediate (human or automatic) phonetic transcription.

The classical pipeline to build state of the art systems for speech recognition consists in first training an HMM/GMM model to force align the units on which the final acoustic model operates (most often context-dependent phone states). This approach takes its roots in HMM/GMM training (Woodland & Young, 1993). The improvements brought by deep neural networks (DNNs) (Mohamed et al., 2012; Hinton et al., 2012) and convolutional neural networks (CNNs) (Sercu et al., 2016; Soltau et al., 2014) for acoustic modeling only extend this training pipeline. The current state of the art on LibriSpeech belongs to this approach too (Panayotov et al., 2015; Peddinti et al., 2015b), with an additional step of speaker adaptation (Saon et al., 2013; Peddinti et al., 2015a). Recently, Senior et al. (2014) proposed GMM-free training, but the approach still requires to generate a forced alignment.

An approach that cut ties with the HMM/GMM pipeline (and with forced alignment) was to train with a recurrent neural network (RNN) (Graves et al., 2013) for phoneme transcription. There are now competitive end-to-end approaches of acoustic models toppled with RNNs layers as in (Hannun et al., 2014; Miao et al., 2015; Saon et al., 2015; Amodei et al., 2016), trained with a sequence criterion (Graves et al., 2006). However these models are computationally expensive, and thus often take a long time to train. On conversational speech (that is not the topic of this paper), the state of the art is still held by complex ConvNets+RNNs acoustic models, coupled to domain-adapted language models (Xiong et al., 2017; Saon et al., 2017).

Compared to classical approaches that need phonetic annotation (often derived from a phonetic dictionary, rules, and generative training), we propose to train the model end-to-end, using graphemes directly. Compared to sequence criterion based approaches that train directly from speech signal to graphemes (Miao et al., 2015), we propose an RNN-free architecture based on convolutional networks for the acoustic model, toppled with a simple sequence-level variant of CTC.

We reach the clean speech performance of (Peddinti et al., 2015b), but without performing speaker adaptation. Our word-error-rate on clean speech is better than (Amodei et al., 2016), while being worse on noisy speech, but they train on 11,900 hours while we only train on the 960h available in LibriSpeech's train set. The rest of the paper is structured as follows: the next section presents the convolutional networks used for acoustic modeling, along with the automatic segmentation criterion and decoding approaches. The last section shows experimental results on LibriSpeech.

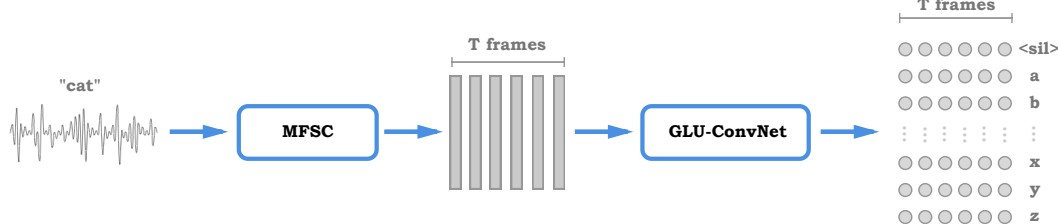

Figure 1: Overview of our acoustic model, which computes MFSC features which are fed to a Gated ConvNet. The ConvNet output one score for each letter in the dictionary, and for each MFSC frame. At inference time, theses scores are fed to a decoder (see Section 2.4) to form the most likely sequence of words. At training time, the scores are fed to the ASG criterion (see Figure 2) which promotes sequences of letters leading to the transcrition sequence (here "c a t").

## 2 ARCHITECTURE

Our acoustic model (see an overview in Figure 1) is a Convolutional Neural Network (ConvNet) (LeCun & Bengio, 1995), with Gated Linear Units (GLUs) (Dauphin et al., 2017). The model is fed with 40 MFSCs features, and is trained with a variant of the Connectionist Temporal Classification (CTC) criterion (Graves et al., 2006), which does not have blank labels but embarks a simple duration model through letter transitions scores (Collobert et al., 2016). During training, we use dropout on the neural network outputs. At inference, the acoustic model is coupled with a decoder which performs a beam search, constrained with a count-based language model. We detail each of these components in the following.

### 2.1 MFSC FEATURES

Our system relies on Mel-Frequency Spectral Coefficients (MFSCs), which are obtained by averaging spectrogram values with mel-scale filters. MFSCs are the step preceding the cosine transform required to compute Mel-Frequency Cepstrum Coefficients (MFCCs), often found in classical HMM/GMM speech systems (Woodland & Young, 1993) because of their dimensionality compression (13 coefficients are often enough to span speech frequencies). Compared to spectrogram coefficients, MFSCs have the advantage to be more robust to small time-warping deformations.

### 2.2 GATED CONVNETS FOR ACOUSTIC MODELING

Our acoustic model is fed with the MFSC frames, and output letter scores for each input frame. At each time step, there is one score per letter in a given dictionary $\mathcal{L}$. Words are separated by a special letter `<sil>`.

The acoustic model architecture is based on a 1D Gated Convolutional Neural Network (Gated ConvNet) (Dauphin et al., 2017). Gated ConvNets stack 1D convolutions with Gated Linear Units. More formally, given an input sequence $\mathbf{X} \in \mathbb{R}^{T \times d^i}$ with $T$ frames of $d$-dimensional vectors, the $i^{\text{th}}$ layer of our network performs the following computation:

$$h^i(\mathbf{X}) = (\mathbf{X} * \mathbf{W}^i + \mathbf{b}^i) \otimes \sigma(\mathbf{X} * \mathbf{V}^i + \mathbf{c}^i), \tag{1}$$

where $*$ is the convolution operator, $\mathbf{W}^i$, $\mathbf{V}^i \in \mathbb{R}^{d^{i+1} \times d^i \times k^i}$ and $\mathbf{b}, \mathbf{c} \in \mathbb{R}^{d^{i+1}}$ are the learned parameters (with convolution kernel size $k^i$), $\sigma(\cdot)$ is the sigmoid function and $\otimes$ is the element-wise product between matrices.

Gated ConvNets have been shown to reduce the vanishing gradient problem, as they provide a linear path for the gradients while retaining non-linear capabilities, leading to state-of-the-art performance both for natural language modeling and machine translation tasks (Dauphin et al., 2017; Gehring et al., 2017).

#### 2.2.1 FEATURE NORMALIZATION AND ZERO-PADDING

Each MFSC input sequence is normalized with mean 0 and variance 1. Given an input sequence $\mathbf{X} \in \mathbb{R}^{T \times d}$, a convolution with kernel size $k$ will output $T - k + 1$ frames, due to border effects.

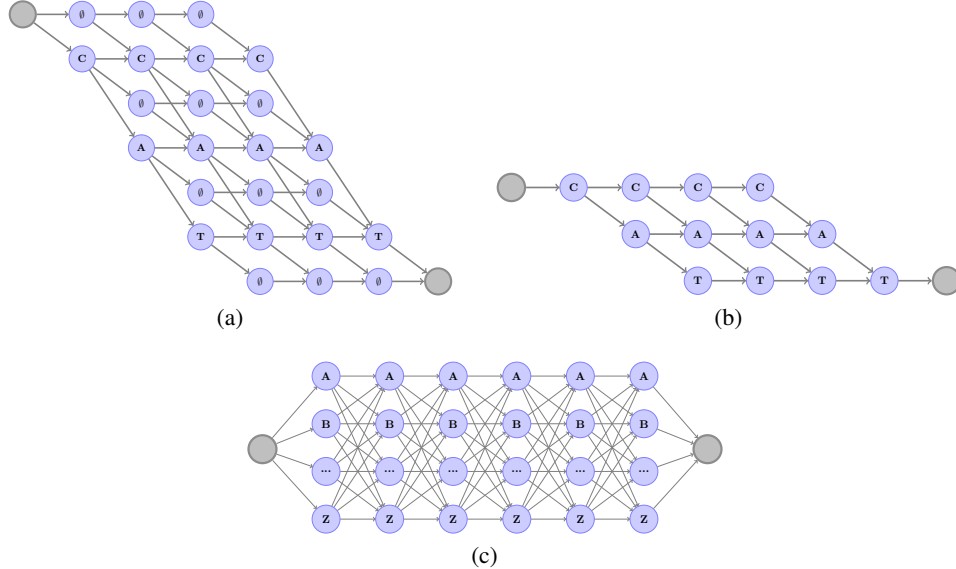

Figure 2: (a) The CTC graph which represents all the acceptable sequences of letters for the transcription "cat" over 6 frames. (b) The same graph used by ASG, where blank labels have been discarded. (c) The fully connected graph describing all possible sequences of letter; this graph is used for normalization purposes in ASG. Un-normalized transitions scores are possible on edges of these graphs. At each time step, nodes are assigned a conditional un-normalized score, output by the Gated ConvNet acoustic model.

To compensate those border effects, we pad the MFSC features $\mathbf{X}^0$ with zeroed frames. To take in account the whole network, the padding size is $\sum_i (k^i - 1)$, divided in two equal parts at the beginning and the end of the sequence.

### 2.3 ACOUSTIC MODEL TRAINING

Most large labeled speech databases provide only a text transcription for each audio file. In a classification framework (and given our acoustic model produces letter predictions), one would need the segmentation of each letter in the transcription to train properly the model. Manually labeling the segmentation of each letter would be tedious. Several solutions have been explored in the speech community to alleviate this issue:

1. HMM/GMM models use an iterative EM procedure: during the Estimation step, the best segmentation is inferred according to the current model, during the Maximization step the model is optimized using the current inferred segmentation. This approach is also often used to boostrap the training of neural network-based acoustic models.

2. In the context of hybrid HMM/NN systems, the MMI criterion (Bahl et al., 1986) maximizes the mutual information between the acoustic sequence and word sequences or the Minimum Bayes Risk (MBR) criterion (Gibson & Hain, 2006). Recent state-of-the-art systems leverage the MMI criterion (Povey et al., 2016).

3. Standalone neural network architectures have also been trained using the Connectionist Temporal Classification (CTC), which jointly infers the segmentation of the transcription while increase the overall score of the right transcription (Graves et al., 2006). In (Amodei et al., 2016) it has been shown that letter-based acoustic models trained with CTC could compete with existing phone-based systems, assuming enough training data is provided.

In this paper, we chose a variant of the Connectionist Temporal Classification. CTC considers all possible sequence sub-word units (e.g. letters), which can lead to the correct transcription. It also allow a special "blank" state to be optionally inserted between each sub-word unit. The rational behind the blank state is two-folds: (i) modeling "garbage" frames which might occur between

each letter and (ii) identifying the separation between two identical consecutive sub-word unit in a transcription. Figure 2a shows the CTC graph describing all the possible sequences of letters leading to the word "cat", over 6 frames. We denote $\mathcal{G}_{ctc}(\theta, T)$ the CTC acceptance graph over $T$ frames for a given transcription $\theta$, and $\pi = \pi_1, \ldots, \pi_T \in \mathcal{G}_{ctc}(\theta, T)$ a path in this graph representing a (valid) sequence of letters for this transcription. CTC assumes that the network output probability scores, normalized at the frame level. At each time step $t$, each node of the graph is assigned with its corresponding log-probability letter $i$ (that we denote $f_i^t(\mathbf{X})$) output by the acoustic model (given an acoustic sequence $\mathbf{X}$). CTC minimizes the Forward score over the graph $\mathcal{G}_{ctc}(\theta, T)$:

$$CTC(\theta, T) = - \operatorname*{logadd}_{\pi \in \mathcal{G}_{ctc}(\theta, T)} \sum_{t=1}^{T} f_{\pi_t}^t(\mathbf{X}), \tag{2}$$

where the "logadd" operation (also called "log-sum-exp") is defined as $\operatorname{logadd}(a, b) = \log(\exp(a) + \exp(b))$. This overall score can be efficiently computed with the Forward algorithm.

### 2.3.1 THE ASG CRITERION

Blank labels introduce complexity when decoding letters into words. Indeed, with blank labels "ø", a word gets many entries in the sub-word unit transcription dictionary (e.g. the word "cat" can be represented as "c a t", "c ø a t", "c ø ø a t", "c ø a ø t", etc... – instead of only "c a t"). We replace the blank label by special letters modeling repetitions of preceding letters. For example "caterpillar" can be written as "caterpil1ar", where "1" is a label to represent one repetition of the previous letter.

Removing blank labels from the CTC acceptance graph $\mathcal{G}_{ctc}(\theta, T)$ (shown in Figure 2a) leads to a simpler graph that we denote $\mathcal{G}_{asg}(\theta, T)$ (shown in Figure 2b). Unfortunately, in practice we observed that most models do not train with this simplification of CTC. Adding *unnormalized* transition scores $g_{i,j}(\cdot)$ on each edge of the graph, when moving from label $i$ to label $j$ fix the issue. We observed in practice that *normalized* transitions led to similar issue that not having transitions. Considering unnormalized transition scores implies implementing a sequence-level normalization, to avoid the model to diverge (represented by the graph $\mathcal{G}_{asg}(\theta, T)$, as shown in Figure 2c). This leads to the following criterion, dubbed ASG for "Auto SeGmentation":

$$ASG(\theta, T) = - \operatorname*{logadd}_{\pi \in \mathcal{G}_{asg}(\theta, T)} \sum_{t=1}^{T} (f_{\pi_t}^t(\mathbf{X}) + g_{\pi_{t-1}, \pi_t}(\mathbf{X})) + \operatorname*{logadd}_{\pi \in \mathcal{G}_{full}(\theta, T)} \sum_{t=1}^{T} (f_{\pi_t}^t(\mathbf{X}) + g_{\pi_{t-1}, \pi_t}(\mathbf{X})). \tag{3}$$

The left-hand part in Equation (3) promotes the score of sequences letters leading to the right transcription (as in Equation (2) for CTC), and the right-hand part demotes the score of all sequences of letters (as does the frame-level normalization – that is the softmax on the acoustic model – for CTC). As for CTC, these two parts can be efficiently computed with the Forward algorithm. When removing transitions in Equation (3), the sequence-level normalization becomes equivalent to the frame-level normalization and the ASG criterion is mathematically equivalent to CTC with no blank labels.

### 2.3.2 OTHER TRAINING CONSIDERATIONS

We apply dropout at the output to all layers of the acoustic model. Dropout retains each output with a probability $p$, by applying a multiplication with a Bernoulli random variable taking value $1/p$ with probability $p$ and $0$ otherwise (Srivastava et al., 2014).

Following the original implementation of Gated ConvNets (Dauphin et al., 2017), we found that using both weight normalization (Salimans & Kingma, 2016) and gradient clipping (Pascanu et al., 2013) were speeding up training convergence. The clipping we implemented performs:

$$\widetilde{\nabla} C = \max(||\nabla C||, \epsilon) \frac{\nabla C}{||\nabla C||}, \tag{4}$$

where $C$ is either the CTC or ASG criterion, and $\epsilon$ is some hyper-parameters which controls the maximum amplitude of the gradients.

Table 1: Architectures details. "#conv." is the number of convolutional layers. Dropout amplitude, "#hu" (number of output hidden units) and "kw" (convolution kernel width) are provided for the first and last layer (all are linearly increased with layer depth). The size of the final layer is also provided.

| Architecture | #conv. | dropout first layer | dropout last layer | #hu first layer | #hu last layer | kw first layer | kw last layer | #hu full connect |
|---|---|---|---|---|---|---|---|---|
| Low Dropout | 17 | 0.25 | 0.25 | 200 | 750 | 13 | 27 | 1500 |
| High Dropout | 19 | 0.20 | 0.60 | 200 | 1000 | 13 | 29 | 2000 |

## 2.4 BEAM-SEARCH DECODER

We wrote our own one-pass decoder, which performs a simple beam-search with beam threholding, histogram pruning and language model smearing Steinbiss et al. (1994). We kept the decoder as simple as possible (under 1000 lines of C code). We did not implement any sort of model adaptation before decoding, nor any word graph rescoring. Our decoder relies on KenLM Heafield et al. (2013) for the language modeling part. It also accepts unnormalized acoustic scores (transitions and emissions from the acoustic model) as input. The decoder attempts to maximize the following:

$$\mathcal{L}(\theta) = \operatorname*{logadd}_{\pi \in \mathcal{G}_{asg}(\theta, T)} \sum_{t=1}^{T} (f_{\pi_t}(x) + g_{\pi_{t-1}, \pi_t}(x)) + \alpha \log P_{lm}(\theta) + \beta |\theta| + \gamma |\{i | \pi_i = \texttt{<sil>}\}|, \quad (5)$$

where $P_{lm}(\theta)$ is the probability of the language model given a transcription $\theta$, $\alpha$, $\beta$, and $\gamma$ are three hyper-parameters which control the weight of the language model, the word insertion penalty, and the silence insertion penalty, respectively.

The beam of the decoder tracks paths with highest scores according to Equation (5), by bookkeeping pair of (language model, lexicon) states, as it goes through time. The language model state corresponds the $(n-1)$-gram history of the $n$-gram language model, while the lexicon state is the sub-word unit position in the current word hypothesis. To maintain diversity in the beam, paths with identical (language model, lexicon) states are merged. Note that traditional decoders combine the scores of the merge paths with a $\max(\cdot)$ operation (as in a Viterbi beam-search) – which would correspond to a $\max(\cdot)$ operation in Equation (5) instead of $\operatorname{logadd}(\cdot)$. We consider instead the $\operatorname{logadd}(\cdot)$ operation, as it takes in account the contribution of all the paths leading to the same transcription, in the same way we do during training (see Equation (3)). In Section 3.1, we show that this leads to better accuracy in practice.

## 3 EXPERIMENTS

We benchmarked our system on LibriSpeech, a large speech database freely available for download (Panayotov et al., 2015). We kept the original 16 kHz sampling rate. We considered the two available setups in LibriSpeech: CLEAN data and OTHER. We picked all the available data (about 960h of audio files) for training, and the available development sets (both for CLEAN, and OTHER) for tuning all the hyper-parameters of our system. Test sets were used only for the final evaluations.

The letter vocabulary $\mathcal{L}$ contains 30 graphemes: the standard English alphabet plus the apostrophe, silence (<SIL>), and two special "repetition" graphemes which encode the duplication (once or twice) of the previous letter (see Section 2.3.1). Decoding is achieved with our own decoder (see Section 2.4), with the standard 4-gram language model provided with LibriSpeech[1], which contains $200,000$ words. In the following, we either report letter-error-rates (LERs) or word-error-rates (WERs).

MFSC features are computed with 40 coefficients, a 25 ms sliding window and 10 ms stride.

We implemented everything using TORCH7[2]. The ASG criterion as well as the decoder were implemented in C (and then interfaced into TORCH).

---

[1] http://www.openslr.org/11.
[2] http://www.torch.ch.

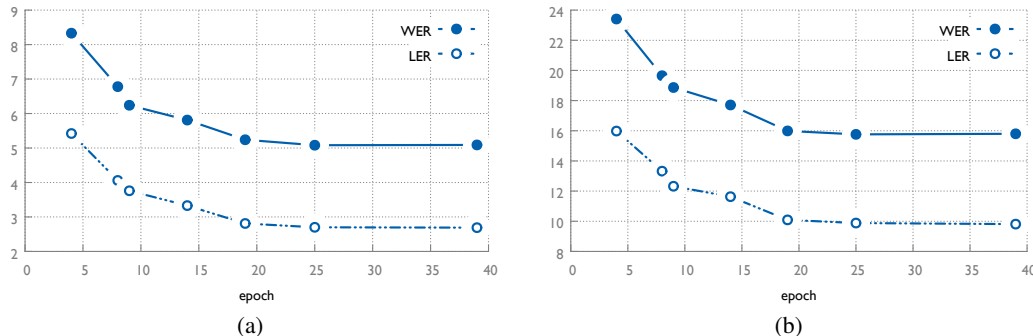

(a)                (b)

Figure 3: LibriSpeech Letter Error Rate (LER) and Word Error Rate (WER) for the first training epochs of our LOW DROPOUT architecture. (a) is on `dev-clean`, (b) on `dev-other`.

Table 2: Comparison in LER and WER of variants of our model on LibriSpeech. When not specified, decoding is performed with the $\mathrm{logadd}(\cdot)$ operation to aggregate similar hypothesis (see Section 2.4).

| model | dev-clean | | dev-other | |
|---|---|---|---|---|
| | LER | WER | LER | WER |
| LOW DROPOUT ($p = 0.2$) | 2.7 | 4.8 | 9.8 | 15.2 |
| HIGH DROPOUT ($p = 0.2 \to 0.6$) | 2.3 | 4.6 | 9.0 | 13.8 |
| HIGH DROPOUT + $\max(\cdot)$ decoding | – | 4.7 | – | 14.0 |

## 3.1 ARCHITECTURE

We tuned our acoustic model architectures by grid search, validating on the dev sets. We consider here two architectures, with low and high amount of dropout (see the parameter $p$ in Section 2.3.2). Table 1 reports the details of our architectures. The amount of dropout, number of hidden units, as well as the convolution kernel width are increased linearly with the depth of the neural network. Note that as we use Gated Linear Units (see Section 2.2), each layer is duplicated as stated in Equation (1). Convolutions are followed by a fully connected layer, before the final layer which outputs 30 scores (one for each letter in the dictionary). This leads to about 130M and 208M of trainable parameters for the LOW DROPOUT and HIGH DROPOUT architectures, respectively.

Figure 3 shows the LER and WER on the LibriSpeech development sets, for the first 40 training epochs of our LOW DROPOUT architecture. LER and WER appear surprisingly well correlated, both on the "clean" and "other" version of the dataset.

In Table 2, we report WERs on the LibriSpeech development sets, both for our LOW DROPOUT and HIGH DROPOUT architectures. Increasing dropout regularize the acoustic model in a way which impacts significantly generalization, the effect being stronger on noisy speech. We also report the WER for the decoder ran with the $\max(\cdot)$ operation (instead of $\mathrm{logadd}(\cdot)$ for other results) used to aggregate paths in the beam with identical (language model, lexicon) states. It appears advantageous (as there is no complexity increase in the decoder) to use the $\mathrm{logadd}(\cdot)$ aggregation.

## 3.2 COMPARISON WITH OTHER SYSTEMS

In Table 3, we compare our system with several of the best systems on LibriSpeech reported in the literature. We highlighted the acoustic model architectures, as well as the type of underlying sub-word unit. Note that phone-based acoustic models output in general senomes; senomes are carefully selected through a complicated procedure involving a phonetic-context-based decision tree built from another GMM/HMM system. Phone-based system also require an additional word lexicon which translates words into a sequence of phones. Most systems also perform speaker adaptation; iVectors compute a speaker embedding capturing both speaker and environment information (Xue

Table 3: Comparison of different ASR systems. We report the type of acoustic model used for various systems, as well as the type of sub-word unit. HMM stands for Hidden Markov Model, CNN for ConvNet; when not specified, CNNs are 1D (also called Time-Delay Neural Networks – TDNN – in the literature). pNorm is a particular non-linearity (Waibel, 1989). We also report extra information (besides word transcriptions) which might be used by each system, including speaker adaptation, or any other domain-specific data.

|  | Acoustic Model | Sub-word | Spkr Adapt. | Extra Resources |
|---|---|---|---|---|
| (Panayotov et al., 2015) | HMM+DNN+pNorm | phone | fMLLR | phone lexicon |
| (Amodei et al., 2016) | 2D-CNN+RNN | *letter* | *none* | 11.9Kh train set, Common Crawl LM |
| (Peddinti et al., 2015b) | HMM+CNN | phone | iVectors | phone lexicon |
| (Povey et al., 2016) | HMM+CNN | phone | iVectors | phone lexicon, phone LM, data augm. |
| (Ko et al., 2015) | HMM+CNN+pNorm | phone | iVectors | phone lexicon, data augm. |
| this paper | GLU-CNN | *letter* | *none* | *none* |

Table 4: Comparison in WER of our model with other systems on LibriSpeech.

|  | test-clean | test-other |
|---|---|---|
| (Panayotov et al., 2015) | 5.5 | 14.0 |
| (Amodei et al., 2016) | 5.3 | 13.3 |
| (Peddinti et al., 2015b) | 4.8 | - |
| (Povey et al., 2016) | 4.3 | - |
| (Ko et al., 2015) | - | 12.5 |
| this paper | 4.8 | 14.5 |
| this paper (*no decoder*) | 6.7 | 20.8 |

et al., 2014), while fMMLR is a two-pass decoder technique which computes a speaker transform in the first pass (Gales & Woodland, 1996).

DEEP SPEECH 2 (Amodei et al., 2016) is the system which is the most related to ours. In contrast to other systems which combine a Hidden Markov Model (HMM) with a ConvNet, DEEP SPEECH 2 is a standalone neural network. In contrast to our system, DEEP SPEECH 2 embarks a more complicated acoustic model composed of a ConvNet and a Recurrent Neural Network (RNN), while our system is a simple ConvNet. Both *Deep Speech 2* and our system rely on letters for acoustic modeling, alleviating the need of a phone-based word lexicon. DEEP SPEECH 2 relies on a lot of speech data (combined with a very large 5-gram language model) to make the letter-base approach competitive , while we limited ourselves to the available data in the LibriSpeech benchmark.

In Table 4, we report a comparison in WER performance for all systems introduced in Table 3. Our system is very competitive with existing approaches. DEEP SPEECH 2 – which is also a letter-based system – is outperformed on clean data, even though our system has been trained with an order of magnitude less data. We report also the WER with no decoder, that is taking the raw output of the neural network, with no alterations. The Gated ConvNet appears very good at modeling true words.

Using a single GPU (no batching), our HIGH DROPOUT Gated ConvNet goes over the CLEAN (5.4h) and OTHER (5.1h) test sets in 4min26s and 4min43s, respectively. The decoder runs over the CLEAN and OTHER sets in 3min56s and 30min5s, using only one CPU thread – which (considering the decoder alone) corresponds to a .01 and 0.1 Real Time Factor (RTF), respectively.

## 4 CONCLUSION

We have introduced a simple end-to-end automatic speech recognition system, which combines a large (208M parameters) but efficient ConvNet acoustic model, an easy sequence criterion which can infer the segmentation, and a simple beam-search decoder. The decoding results are competitive on the LibriSpeech corpus (4.8% WER dev-clean). Our approach breaks free from HMM/GMM

pre-training and forced alignment, as well as not being as computationally intensive as RNN-based approaches (Amodei et al., 2016). We based all our work on a publicly available (free) dataset, all of which should make it easier to reproduce. Further work should include leveraging speaker identity, training from the raw waveform, data augmentation, training with more data, better language models.

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
