# OpenReview forum: "Gated ConvNets for Letter-Based ASR"
_ICLR.cc/2018/Conference — Reject_

### Official Review · AnonReviewer3 · 2017-11-15
**Minimal novelty**

**Rating:** 3
**Confidence:** 5

**Review:**

This paper applies gated convolutional neural networks [1] to speech recognition, using the training criterion ASG [2]. It is fair to say that this paper contains almost no novelty.

This paper starts by bashing the complexity of conventional HMM systems, and states the benefits of their approach. However, all of the other grapheme-based end-to-end systems enjoy the same benefit as CTC and ASG. Prior work along this line includes [3, 4, 5, 6, 7].

Using MFSC, or more commonly known as log mel filter bank outputs, has been pretty common since [8]. Having a separate subsection (2.1) discussing this seems unnecessary.

Arguments in section 2.3 are weak because, again, all other grapheme-based end-to-end systems have the same benefit as CTC and ASG. It is unclear why discriminative training, such as MMI, sMBR, and lattice-free MMI, is mentioned in section 2.3. Discriminative training is not invented to overcome the lack of manual segmentations, and is equally applicable to the case where we have manual segmentations.

The authors argue that ASG is better than CTC in section 2.3.1 because it does not use the blank symbol and can be faster during decoding. However, once the transition scores are introduced in ASG, the search space becomes quadratic in the number of characters, while CTC is still linear in the number characters. In addition, ASG requires additional forward-backward computation for computing the partition function (second term in eq 3). There is no reason to believe that ASG can be faster than CTC in both training and decoding.

The connection between ASG, CTC, and marginal log loss has been addressed in [9], and it does make sense to train ASG with the partition function. Otherwise, the objective won't be a proper probability distribution.

The citation style in section 2.4 seems off. Also see [4] for a great description of how beam search is done in CTC.

Details about training, such as the optimizer, step size, and batch size, are missing. Does no batching (in section 3.2) means a batch size of one utterance?

In the last paragraph of section 3.2, why is there a huge difference in real-time factors between the clean and other set? Something is wrong unless the authors are using different beam widths in the two settings.

The paper can be significantly improved if the authors compare the performance and decoding speed against CTC with the same gated convnet. It would be even better to compare CTC and ASG to seq2seq-based models with the same gated convnet. Similar experiments should be conducted on switchboard and wsj because librespeech is several times larger than switchboard and wsj. None of the comparison in table 4 is really meaningful, because none of the other systems have parameters as many as 19 layers of convolution. Why does CTC fail when trained without the blanks? Is there a way to fix it besides using ASG? It is also unclear why speaker-adaptive training is not needed. At which layer do the features become speaker invariant? Can the system improve further if speaker-adaptive features are used instead of log mels? This paper would be much stronger if the authors can include these experiments and analyses.

[1] R Collobert, C Puhrsch, G Synnaeve, Wav2letter: an end-to-end convnet-based speech recognition system, 2016

[2] Y Dauphin, A Fan, M Auli, D Grangier, Language modeling with gated convolutional nets, 2017

[3] A Graves and N Jaitly, Towards End-to-End Speech Recognition with Recurrent Neural Networks, 2014

[4] A Maas, Z Xie, D Jurafsky, A Ng, Lexicon-Free Conversational Speech Recognition with Neural Networks, 2015

[5] Y Miao, M Gowayyed, F Metze, EESEN: End-to-end speech recognition using deep RNN models and WFST-based decoding, 2015

[6] D Bahdanau, J Chorowski, D Serdyuk, P Brakel, Y Bengio, End-to-end attention-based large vocabulary speech recognition, 2016

[7] W Chan, N Jaitly, Q Le, O Vinyals, Listen, attend and spell, 2015

[8] A Graves, A Mohamed, G Hinton, Speech recognition with deep recurrent neural networks, 2013

[9] H Tang, L Lu, L Kong, K Gimpel, K Livescu, C Dyer, N Smith, S Renals, End-to-End Neural Segmental Models for Speech Recognition, 2017

---

### Official Review · AnonReviewer1 · 2017-11-22
**Interesting- maybe as workshop paper**

**Rating:** 6
**Confidence:** 5

**Review:**

The paper describes some interesting work but for a combination of reasons I think it's more like a workshop-track paper.
There is not much that's technically new in the paper-- at least not much that's really understandable.   There is some text about a variant of CTC, but it does not explain very clearly what was done or what the motivation was.
There are also quite a few misspellings.
Since the system is presented without any comparisons to alternatives for any of the individual components, it doesn't really shed any light on the significance of the various modeling decisions that were made.  That limits the value.
If rejected from here, it could perhaps be submitted as an ICASSP or Interspeech paper.

---

### Official Review · AnonReviewer2 · 2017-11-27
**More work needed**

**Rating:** 4
**Confidence:** 4

**Review:**

The paper is interesting, but needs more work, and should provide clear and fair comparisons. Per se, the model is incrementally new, but it is not clear what the strengths are, and the presentations needs to be done more carefully.

In detail:
- please fix several typos throughout the manuscript, and have a native speaker (and preferably an ASR expert) proofread the paper

Introduction
- please define HMM/GMM model (and other abbreviations that will be introduced later), it cannot be assumed that the reader is familiar with all of them ("ASG" is used before it is defined, ...)
- The standard units that most ASR systems use can be called "senones", and they are context dependent sub-phonetic units (see http://ssli.ee.washington.edu/~mhwang/), not phonetic states. Also the units that generate the alignment and the units that are trained on an alignment can be different (I can use a system with 10000 states to write alignments for a system with 3000 states) - this needs to be corrected.
- When introducing CNNs, please also cite Waibel and TDNNs - they are *the same* as 1-d CNNs, and predate them. They have been extended to 2-d later on (Spatio-temporal TDNNs)
- The most influential deep learning paper here might be Seide, Li, Yu Interspeech 2011 on CD-DNN-HMMs, rather than overview articles
- Many papers get rid of the HMM pipeline, I would add https://arxiv.org/abs/1408.2873, which predates Deep Speech
- What is a "sequence-level variant of CTC"? CTC is a sequence training criterion
- The reason that Deep Speech 2 is better on noisy test sets is not only the fact they trained on more data, but they also trained on "noisy" (matched) data
- how is this an end-to-end approach if you are using an n-gram language model for decoding?

Architecture
- MFSC are log Filterbanks ...
- 1D CNNs would be TDNNs
- Figure 2: can you plot the various transition types (normalized, un-normalized, ...) in the plots? not sure if it would help, but it might
- Maybe provide a reference for HMM/GMM and EM (forward backward training)
- MMI was also widely used in HMM/GMM systems, not just NN systems
- the "blank" states do *not* model "garbage" frames, if one wants to interpret them, they might be said to model "non-stationary" frames between CTC "peaks", but these are different from silence, garbage, noise, ...
- what is the relationship of the presented ASG criterion to MMI? the form of equation (3) looks like an MMI criterion to me?

Experiments
- Many of the previous comments still hold, please proofread
- you say there is no "complexity" incrase when using "logadd" - how do you measure this? number of operations? is there an implementation of "logadd" that is (absolutely) as fast as "add"?
- There is discussion as to what i-vectors model (speaker or environment information) - I would leave out this discussion entirely here, it is enough to mention that other systems use adaptation, and maybe re-run an unadapted baselien for comparsion
- There are techniques for incremental adaptation and a constrained MLLR (feature adaptation) approaches that are very eficient, if one wnats to get into this
- it may also be interesting to discuss the role of the language model to see which factors influence system performance
- some of the other papers might use data augmentation, which would increase noise robustness (did not check, but this might explain some of the results in table 4)
- I am confused by the references in the caption of Table 3 - surely the Waibel reference is meant to be for TDNNs (and should appear earlier in the paper), while p-norm came later (Povey used it first for ASR, I think) and is related to Maxout
- can you also compare the training times?

Conculsion
- can you show how your approach is not so computationally expensive as RNN based approaches? either in terms of FLOPS or measured times

---

### Public Comment · (anonymous) · 2017-11-13
**Notorious comment on missing citations**

The paper seems to completely ignore a set of works on character-based ASR with attention networks:
  - Chorowski et al. "Attention-based models for speech recognition.", 2015
  - Chan et al. "Listen, attend and spell", 2015
  - Bahdanau et al. "End-to-end attention-based large vocabulary speech recognition.", 2016
and some milestone works with CTC loss function:
  - Graves and Jaitly "Towards end-to-end speech recognition with recurrent neural networks." , 2014
  - Zhang et al. "Towards end-to-end speech recognition with deep convolutional neural networks.", 2017

This work uses rather unusual corpus LibriSpeech, therefore the performance is not comparable to works listed above (that benchmark mainly on WSJ). LibriSpeech is great, but the paper lacks comparison to prior work on end-to-end recognition.

MFSC features are presented as an invention in this paper. Such features are usually referred as "log-mel filterbank" in the literature.

---

### Author Response · Authors · 2017-12-13
**reply to reviewers**

General comments:

The point of the paper is that letter-based systems can compete with phone/senone-based systems, with no extra training data. We will clarify that letter-based systems (also called grapheme-based systems) predate all the recommended citations (see "Context-dependent acoustic modeling using graphemes for large vocabulary speech recognition", 2002 or "Grapheme based speech recognition", 2003). However, previous letter-based work report WER far behind from phone-based systems.

Concerning recommended citations: TIMIT ones are not very relevant, as they report phone error rate (no WER). TIMIT is also a tiny dataset. WSJ-related citations are far from reaching SOTA — and WSJ is an order of magnitude smaller than LibriSpeech. We will add:
- https://arxiv.org/pdf/1508.01211.pdf: closer than other work to SOTA, and comparable to LibriSpeech in size — not reproducible though (Google data).
- https://arxiv.org/pdf/1708.00531.pdf: compares ASG from a formal standpoint.

We never claimed MFSC are novel. We will keep their descriptions for persons less familiar with speech features. We will switch to the more common name "log mel-filterbanks".

Reviewer 1:

- how is this an end-to-end approach if you are using an n-gram language model for decoding?

Same concept of end-to-end than other existing approaches (e.g. Deep Speech 2 uses an n-gram LM).

- MMI was also widely used in HMM/GMM systems, not just NN systems

Indeed - we will reword.

- what is the relationship of the presented ASG criterion to MMI? [...]

ASG can be viewed as related to a MMI criterion with a letter-based LM. However, ASG uses a discriminative model P(label|sound) = P(Y|X) instead of P(X|Y) in MMI. This goes out of the scope of this paper, though.

- you say there is no "complexity" incrase when using "logadd" [...]

We meant code complexity (one only needs to replace "max" by "logadd") - we will fix.

- There is discussion as to what i-vectors model [...]

We will add a Kaldi baseline with no speaker adapation.

- I am confused by the references in the caption of Table 3 [...]

Indeed - we will fix.

- can you also compare the training times?

Hard to do so without ending up comparing "implementations".

Reviewer 2:

- There is not much that's technically new in the paper [...]  There is some text about a variant of CTC [...]

We are working on a version of Table 2 with CTC-trained models. CTC and ASG leads to similar results in our experience; the advantage of ASG is that there is no blank state, which simplifies the decoder implementation.

Reviewer 3:

- It is fair to say that this paper contains almost no novelty.

This is the first paper to show that letter-based systems can reach phone/senone-based systems performance, on a standard dataset, with no additional data.

- This paper starts by bashing the complexity of conventional HMM systems [...]

We do not bash! We cite and explain the lineage of speech recognition systems.

- It is unclear why discriminative training, such as MMI [...], is mentioned

"discriminative" is not present in our paper, we mention these criterions as they relate to CTC and ASG.

- The authors argue that ASG is better than CTC [...] because it does not use the blank symbol and can be faster during decoding.

We argue that ASG (without blank labels) makes the decoder's code simpler, not computationally more efficient.

- [...] in ASG, the search space becomes quadratic in the number of characters, while CTC is still linear [...]

It's quadratic in the # of characters in the dictionary, linear in sequence length. CTC is linear in both (dict size/sentence length): it is faster for languages with a large # of characters (Chinese). For English the runtime is dominated by sequence length anyway.

- The citation style in section 2.4 seems off.

We will fix.

- Details about training [...] are missing. Does no batching [...]

We will fix; yes: batch = 1 utterance in section 3.2.

- [...] why is there a huge difference in real-time factors between the clean and other set? Something is wrong unless [...]

There is nothing wrong, the decoder has a score-based limit on the beam width, and noisy speech produces way larger beams.

- The paper can be significantly improved if the authors compare the performance and decoding speed against CTC with the same gated convnet

We are working on it.

- None of the comparison in table 4 is really meaningful [...]

All systems are trained on LibriSpeech and without limitations, our comparison is as meaningful as it gets.

- It is also unclear why speaker-adaptive training is not needed

We did not say it is not needed; it is likely that speaker adaptation helps to reduce WER even further (future work).

- At which layer do the features become speaker invariant?

It gets very hard to classify speakers (and thus speaker invariant) after the first few layers. (part of future work).

- Can the system improve further if speaker-adaptive [...]?

Possibly (future work).

---

> ### Comment · AnonReviewer3 · 2017-12-16
> **minor comments**
>
> Thanks for the reply.
>
> > ASG can be viewed as related to a MMI criterion with a letter-based LM. However, ASG uses a discriminative model P(label|sound) = P(Y|X) instead of P(X|Y) in MMI. This goes out of the scope of this paper, though.
>
> I hope you are just misremembering. MMI is maximizing P(Y|X). See [1, 2]. MMI is a general loss function, and need not be tied to lattices and HMMs. The only minor difference between MMI and ASG is whether the segmentations are marginalized. Sometimes people marginalize segmentations when using MMI, e.g., in [2]. In this case, MMI and ASG are equivalent.
>
> > It's quadratic in the # of characters in the dictionary, linear in sequence length. CTC is linear in both (dict size/sentence length): it is faster for languages with a large # of characters (Chinese). For English the runtime is dominated by sequence length anyway.
>
> This should be noted in the paper. CTC can be applied to label sets beyond characters [3, 4], so the dependency on the size of the label set matters.
>
> > This is the first paper to show that letter-based systems can reach phone/senone-based systems performance, on a standard dataset, with no additional data.
>
> > All systems are trained on LibriSpeech and without limitations, our comparison is as meaningful as it gets.
>
> I agree with this statement and I agree the authors have detailed the how, but the more important question is why. Is it because of ASG? Is it because of switching from plain CNNs to gated CNNs? Or is it just because of better tuning? Building a system with known techniques and better tuning should not be considered as a contribution. This is also why I said the comparison in table 4 is meaningless. The authors should at least conduct experiments to show where the improvements are coming from. Please at least compare against CTC, and at least compare plain CNNs against gated CNNs. It would be even better if the authors can use the same network architectures as the other papers appeared in table 4 to compare CTC and ASG. Having the right control experiments is the very basic of a scientific study.
>
> [1] L Bahl, P Brown, P de Souza, R Mercer, Maximum Mutual Information Estimation of Hidden Markov Model Parameters for Speech Recognition, 1986
>
> [2] D Povey, PC Woodland, Minimum Phone Error and I-Smoothing for Improved Discriminative Training, 2007
>
> [3] H Soltau, H Liao, H Sak, Neural speech recognizer: Acoutic-to-word LSTM model for large vocabulary speech recognition, 2016
>
> [4] H Liu, Z Zhu, X Li, S Satheesh, Gram-CTC: Automatic Unit Selection and Target Decomposition for Sequence Labelling, 2017

---

> > ### Author Response · Authors · 2017-12-18
> > **MMI vs ASG**
> >
> > Sorry for the confusion, what we meant is:
> > ASG computes P(Y_r|X_r) = S(Y_r|X_r)/sum_{Y}S(Y|X_r)
> > MMI computes P(Y_r|X_r) = P(X_r|Y_r)P(Y_r) / (sum_{Y}P(X_r|Y)P(Y))
> > The difference in the conditioning order of Ys and Xs are within the summation. Another difference of ASG with vanilla MMI is that P(Y_r|X_r) is computed using unnormalized scores.

---

### Public Comment · (anonymous) · 2017-12-17
**citations and comparison**

"Concerning recommended citations: TIMIT ones are not very relevant, as they report phone error rate (no WER). TIMIT is also a tiny dataset. WSJ-related citations are far from reaching SOTA — and WSJ is an order of magnitude smaller than LibriSpeech."

TIMIT/WSJ citations should not be missing because "tiny dataset" or "far from reaching SOTA" .
1] Research is progressive, we start off with small problems then move onto bigger ones. We should not discount prior work because they were done on smaller problems or weren't as successful. We should acknowledge and compare to prior work.
2] WSJ is not far from SOTA, see:
Jan Chorowski and Navdeep Jaitly, "Towards better decoding and language model integration in sequence to sequence models", in INTERSPEECH 2017. They achieve near DNN-HMM (when compared w/o speaker adaptation).

Citation missing Alex Grave's paper, this is arguably the paper that kicked off the whole field of end2end ASR.
Alex Graves and Navdeep Jaitly, "Towards End-To-End Speech Recognition with Recurrent Neural Networks" in ICML 2014. This citation is critical and missing.

Citation missing for Hori's work. Their CTC+attention model surpasses DNN-HMM models for both chinese and japanese compared to the Kaldi MMI recipe:
Hori et al., "Advances in Joint CTC-Attention based End-to-End Speech Recognition with a Deep CNN Encoder and RNN-LM", in INTERSPEECH 2017.

The authors also selected a weird dataset librespeech instead of WSJ. The vast majority of prior literature on end2end ASR has been done on WSJ, including CTC and seq2seq. A fair comparison needs to be done to CTC and seq2seq, it is very unclear reading this paper how the model compares to other end2end results (i.e., how would it fair compared to label smoothing, CTC+attention model, Latent Sequence Decomposition; all of which is published on WSJ). There should be no reason to avoid comparing new work to prior work/literature.

Also, one might argue this paper is not end2end as it requires a n-gram LM to get reasonable results. CTC/ASG models do not perform well w/o LM (relative to seq2seq).

---

### Decision · Program_Chairs · 2018-01-29
**ICLR 2018 Conference Acceptance Decision**

**Decision:**

Reject

**Comment:**

Pros
-- Competitive results on LibriSpeech.
Cons
-- Limited novelty, and lacks enough comparisons.
-- Comparison with other end-to-end approaches, and on other commonly used datasets, like WSJ, missing.
-- Gated convnets have already been proposed.
-- Letter based systems have been shown to be competitive to phone based systems.
-- Optimization criterion is quite similar to lattice-free MMI proposed by Povey et al., but with a letter based LM and a slightly different HMM topology.

Given the cons pointed out by reviews, the AC is recommending that the paper be rejected.